# Portland Cements with High Content of Calcined Clay: Mechanical Strength Behaviour and Sulfate Durability

**DOI:** 10.3390/ma13184206

**Published:** 2020-09-22

**Authors:** Carlos H. Aramburo, César Pedrajas, Rafael Talero

**Affiliations:** 1Cement Industry Consultant, Carrera 2A oeste No. 7-92, Cali 760045, Colombia; caramburov@gmail.com; 2R&D, Cementos ARGOS, Carrera 49#7 Sur 50, Medellin 050017, Colombia; 3SACACH S.L., c/Santa Susana 32.4°.2, 28033 Madrid, Spain; rtalero@sacach.com

**Keywords:** calcined clays, pozzolanic addition, low carbon cement, sulfate attack

## Abstract

Calcined clay has become the supplementary cementitious materials with the greatest potential to reduce the clinker/cement. In this research, the mechanical strengths and sulphate resistance of blended cements with a high content of calcined clay as a pozzolanic addition were evaluated to demonstrate that these cements could be designed as CEM (cement) type IV/A-SR and IV/B-SR cements by the current European standard UNE-EN 197-1: 2011. The blended cements were prepared by two Portland cements (P1 and PY6) with different mineralogical compositions and a calcined clay. The level of replacement was greater than 40% by weight. The results obtained confirm the decrease in the mechanical strengths and the increase in the sulfate resistance of the two Portland cements when they are replaced by calcined clay at a level of replacement greater than 40%. These results are a consequence of the chemical effect from the pozzolanic activity of the calcined clay. Therefore, there is an important decrease in portlandite levels of paste liquid phase that causes the increase in sulfate resistance and the decrease of the mechanical strengths.

## 1. Introduction

Portland cement-based materials such as concrete are nowadays the most widely used construction materials. Due to a global production of cement and big quantities of concrete produced and consumed globally, the cement industry accounts for around 4% of the total global greenhouse gas emissions and up to 8% of the total global anthropogenic CO_2_ emissions [1]. The partial replacement of Portland cement clinker by supplementary cementitious materials (SCM) is more and increasingly the focus of cement research. The main reason for this development is the potential to limit the CO_2_ emission associated with the production of Portland cement clinker [2]. Mineral additions have been used in cement production industry for many years. Essentially, two types are ground in many cases with the Portland clinker or blended directly into the cement too: pozzolanic additions (natural and artificial pozzolans, fly ash, silica fume, or hydraulically active mineral additions (slag blast furnace)) and crystalline fillers, which are not pozzolanic, such as limestone.

The greater or lesser pozzolans reactivity on portlandite (Ca(OH)_2_), depends primarily on their amorphism, state, and reactive nature of their structure [3,4,5,6]. The cement industry has also gradually evolved toward the gradual inclusion, along with pozzolans, of crystalline mineral additions known as fillers, some of which interact physically and/or chemically with Portland cement [7].

One of the real and proven alternative sources are calcined clays [8], due to clay minerals being an abundant and widespread material. The production process of calcined clay is less energy intensive due to lower firing temperatures and the absence of a decarbonation reaction. The calcination of clays occurs in the temperature range of 600–850 °C and results in the dihydroxylation of the clay whereby an amorphous phase is formed. The Si and Al in this phase can chemically react at ambient temperatures with Ca(OH)_2_ (portlandite), which is formed during cement hydration, in the presence of water to form compounds that possess cementitious properties. Such pozzolanic activity derives from the clay fraction present in raw material.

According to Talero [9], calcined clays are aluminic pozzolans in chemical character, although ASTM C618-08 standard [10] classifies them as siliceous and aluminous materials in nature. Aluminic pozzolans additions lower the sulfate resistance of the Portland cement with which they are blended (sometimes very rapidly), even where the cement Portland is highly resistant to sulfate [11,12]. In the absence of sulfates, however, they raise the mechanical strength, especially early age strength [13], and prevent or at least hinder chloride attack on the steel in reinforced concrete, first chemically and then physically [14].

The use of high-volume calcined clay as a partial replacement of cement in concrete is being studied. The main concern in this regard is whether cement can be replaced by calcined clay above the limiting quantity of 15–20% by mass in the concrete [15,16,17,18,19,20,21]. Indeed, this percentage is beneficial in optimizing the low cost, but it may not improve the durability to any considerable extent. Now, the environmental benefit that could be achieved by replacing a considerable portion of the cement with calcined clays will depend greatly on the strength and durability performance of the material. The concrete in its expected future natural environment, in this case an environment where carbonation-induced corrosion is a risk, will require regular maintenance, repair, or even full replacement [22].

In addition, sulfates species are the most important natural aggressive agents affecting cement matrices. Consequently, there are many standards that establish chemical specifications to guarantee suitable cement resistance for these ions. In the absence of chemical specification(s), however, accelerated attacks must be simulated, using methods such as the Le Chatelier–Anstett test [23,24].

Both items will be addressed in this paper to provide some information of the potential behaviour of these types of blended cements in sulfate environments.

So, the aim of this research is to demonstrate that these cements could be designed as a CEM type IV / A-SR and IV / B-SR cements by the current European standard UNE-EN 197-1: 2011 [25] and to explain the reason for its sulphate resistance

## 2. Materials and Methods

### 2.1. Bill of Materials

The following materials were selected (Table 1):Two Portland cements (P, Portland Valderrivas company, Madrid, Spain), with an excessively big difference in terms of mineralogical phase composition, were chosen to ensure that the results would be extensive to any type of P. The first one was denominated P1 and characterized by its high C_3_A(%) content. On the other hand, the second Portland cement, PY6, was selected due to its low C_3_A(%) and high C_3_S(%) contents.Calcined clay (M) was prepared by a clay with kaolinite mineral content of around 45%. This material was calcined at 750 °C, its granular composition being in accordance with ASTM C 595M-95 standard [26] (retained at wet sieved on n° 325 (45 µm) sieve, max = 20%). The elemental composition of M was determined by X-ray fluorescence (XRF). In addition, reactive silica content was determined by Spanish standard UNE 80-225 [27] (16.17%) and reactive alumina content by the Florentin [28] procedure (9.67%).Gypsum, natural stone (Ibericos Gypsum S.A., Madrid, Spain) (with a high CaSO_4_ 2H_2_O content, 95.58%) was used as aggressive media. The SO_3_ content of this type of sulfate recourse was 45.87%.Lime, Calcium hydroxide (CH) (Carbonatos calcicos del norte, Madrid, Spain), 98%, extra pure.

### 2.2. Experimental Procedure

The two P were blended (in dry) with M at replacement ratios of 40%, 50%, 60%, and 70%. The 100/00 ratio denotes a pure Portland cement. Two series of blended cements were made: one with sufficient gypsum for SO_3_ to account for 7.0% of the total cementitious material in each sample, to enhancing kinetics system, and the other gypsum-free. The two P were blended (in dry) with M at replacement ratios of 30%, 40%, 50%, 60%, and 70%. In addition, one series of blended cements was prepared with an extra content of lime, replacing part of the calcined clay with calcium hydroxide. Therefore, these blended cements contained a proportion of CH/M 1/6% percentage by weight.

Firstly, the pozzolanity of P1 blended cements were tested by Frattini method EN 196-5 Standard [29], that determines the amount of Ca(OH)_2_ originated from cement hydration that taking place in an aqueous solution containing the sample at several ages (7, 28, 60, and 90 days). These [OH-] and [CaO] results must be compared with the calcium hydroxide [Ca(OH)_2_] solubility isotherm curve estimated for an alkaline solution at 40 °C. Those points underneath that curve (subsaturation zone) are indicative the pozzolanic activity of the material that the cement was blended. The P/M replacement ratios used in this test were 70/30%, 60/40%, 50/50%, and 40/60% by weight.

Later, all blended cemesnts were batched in 60/40%, 50/50%, 40/60%, and 30/70% proportions of Portland cement (%)/‘‘M’’ calcined clay to determine their mechanical strengths values in accordance with EN 196-1 standard [30]. The tests were carried out at 7, 28, and 90 days of age. Additional series of 50/50%, 40/60%, and 30/70% blended cements were prepared with an extra content of calcium hydroxide were tested too. Finally, both mixtures 30/70% were supplemented with 7% additional SO_3_ and their mechanical strengths were determined.

This standard includes the compression and flexural strengths determination of prismatic specimens of dimensions 40 × 40 × 160 mm. Tree specimens of each blended cement are manufactured with a plastic mortar, composed of one part by mass of cement, three parts by mass of CEN standardized sand [30], and half part of water (ratio w/c 0.5). The specimens are kept in the mold in a humid atmosphere for 24 h and, after debonding, the specimens are immersed in water until the moment of resistance tests. At the required age (7, 28, and 90 days for this research), the specimens are removed from their moist preservation medium, they are broken in flexion procedure, the flexural strength determined, and each half is subjected to compression trial.

The result of the flexural strength test for each blended cement is calculated as the arithmetic mean of the three individual results, expressed, each one of them, rounded to the nearest 0.1 MPa and obtained from the determination made on the set of three test specimens. In the same way, the result of the compressive strength test for each blended cement is calculated as the arithmetic mean of the six individual results, expressed, each one, rounded to the nearest 0.1 MPa and obtained from the six determinations made on the set of three specimens.

In addition, each blended cement (P1/M and PY6/M) and both P (P1 and PY6) were submitted to the L–A test [23]. In this accelerated method, every blended cement was ground to pass a sieve with an approximately 150-µm opening (100 number by ASTM standard). The amount of these was 100.0 g, approximately. Them, for all of blended cements, a cement paste was made into a 0.50 water/cement ratio, allowed to harden for several days (commonly two weeks), and subsequently crushed to 5 mm in size and dried at over 40 °C. After this water removing process, every cement paste was blended with gypsum (as 50% of the dried set cement), and the mixture was ground to pass a 200-number sieve with an approximately 75-µm opening to accelerate the internal damage by sulfate attack. Six percent by weight of distilled was added, and the final blended cement paste is placed in a cylindrical mold, 80 × 30 mm high, and compressed under a pressure of 196 Pa/min. The initial diameter of every specimen Ø_0_ = 80 mm. After the manufacturing process, each specimen was stored was stored in a chamber, keeping relative humidity at 100% RH, in separate airtight jars over a film of distilled water. The internal temperature in these jars was held at 21 ± 2 °C. The diameter value was measured at times varying from 1 to 730 days (1, 7, 28, 60, 90, 180, 365, and 730 days). The ΔØ (%) values obtained were compared with the allowed specification at 28 days, 1.25% (physical requirement) [9,24], and they were used to determine the speed of diameter growth, Vcø (ΔØ/day), which was obtained dividing the diameter increase, ΔØ (%), between the days of each measurement.

Finally, a few grams of L–A specimens were collected, dried in a CO_2_-free atmosphere, and ground to the original cement particle size. These L-A samples were analyzed on a Philips PW 1730 X-ray powder diffractor (XRD, PW Philips, Madrid, Spain). The scanning rate was 0.02 °C and the step time 0.80 s in a range of 5–60 °C. The technical machine parameters: a graphite monochromator (power settings of 40 kV and 20 mA) and set for CuK1 = 1.54056 and CuK2 ¼ 1.54439 radiation, (K1/K2) = 0.50.

## 3. Results and Discussion

### 3.1. Pozzolanicity Test

Figure 1 shows the results of the pozzolanicity test. This test implies the determination of the amount of calcium cation (Ca^2+^) and hydroxyl anion (OH^−^) containing in the water in contact with the tested samples at 40 °C. In this case the test was made at 7, 28, 60, and 90 days.

M calcined clay exhibited high reactivity with Portland cement P1 due to the high content in hydraulic factors of this materials (SiO_2_^r−^ (38.00%) and Al_2_O_3_^r−^ (15.00%)). The Ca^2+^ than OH^–^ ions consumption of blended cements was much higher and appeared earlier than PC (in this case that pozzolanic activity pozzolanicity never appears). Moreover, pozzolanic activity grew with the M ratio in blended cements.

This CC used in this research possesses certain pozzolanic properties due to their capacity to react with portlandite which can be seen in the decreasing vs. time [CaO] amount in the liquid phase in the case of all blends. CC tends to sustain the [OH^−^] content and react rapidly with Ca^2+^ because of its respective Al_2_O_3_^r−^ content, forming several hydrated calcium aluminates, particularly Stratling’s compound [13] and, if gypsum is present in the medium, hydrated calcium sulfoaluminates: ett-rf and AFm phase [9,11,12,24]. This phenomenon is explained by the high pozzolanic activity of CC, since the physical state of its SiO_2_^r−^ (%) and Al_2_O_3_^r−^ (%) is completely amorphous [4], thus Frattini test compliance is not surprising.

In summary, it could be said that, as the percentage CC in cement pastes increases, the consumption of Ca(OH)_2_ increases significantly. This behavior is directly related to the content of SiO_2_^r−^ (%) and Al_2_O_3_^r−^ (%) of the CC. Therefore, it is demonstrated that, for a highly pozzolanic cement, if the CC content is increased, there will be less portlandite available to develop the pozzolanic action. So, there will be a moment when the pozzolan substitution level is so high that the cement liquid phase reaches a level of subsaturation so extreme that there isn’t enough portlandite in the medium for the formation of new CSH gels and hydrated calcium aluminates of pozzolan origin. Consequently, mechanical strengths decrease with highly added blended cements. This undesirable behavior will occur in a different percentage of substitution, depending on the SiO_2_^r−^ (%) and Al_2_O_3_^r−^ (%) contents of any pozzolan addition.

### 3.2. Mechanical Strenghts

In this section, the results of mechanical strengths values in accordance with EN 196-1 standard [24] at 7, 28, and 90 days are showed. Blended cements that were analyzed are the following: 60/40%, 50/50%, 40/60%, and 30/70% proportions of Portland cement (%)/M calcined clay, and additional series of 50/50%, 40/60%, and 30/70% blended cements with an extra content of calcium hydroxide and SO_3_, as shown in Table 2.

The flexural and compressive strengths values of blended cements are shown in Figure 2. From the absolute point of view, P/M blended cements demonstrated a good mechanical performance. The 60/40% and 50/50% blended cements reached the minimum compressive strength required in the current European standard EN 197-1 [25] for common cements (32.5 MPa at 28 days). The main reason that supports this behavior the high, fast, and early pozzolanic activity due to the Al_2_O_3_^r−^ (%) content of CC [4]. This phenomenon occurred also, in a similar way, in flexion strength tests whereby the same blended cements showed the best mechanical results. Thus, about the specific case of P1/M 60/40 and PY6/M 60/40 blended cements, their performances were higher than the reference Portland cement. This behavior was faster on P1 due to the synergic effect developed between the high Al_2_O_3_^r−^ content (%) of the calcined clay and the high C_3_A content of the P1 cement [31].

However, there are several results that need to be analyzed separately:Firstly, there was a significant decrease in the mechanical strengths of blended cements when the percentage of substitution by pozzolan was increased. The explanation of this result can be approached from two points of view: first, by the progressive reduction of clinker content in cements when the calcined clay content increased, and secondly, by increasing the content of pozzolan in blended cements. The high pozzolanic activity of M produced a significant reduction of portlandite level in liquid phase of each mortar. Due to this phenomenon, activated clay didn’t develop all its pozzolanic activity to generate hydration products because there was not enough Ca(OH)_2_ to react with the pozzolan. This hypothesis was verified when a certain amount of CH was added to each blended cement. By performing this practice, all blended cements increased their mechanical strengths significantly at all ages. Furthermore, blended cements P/M 60/40 confirm that this ratio was the only one that had enough portlandite, so CC could develop a complete pozzolanic activity. The mechanical performance of these cements exceeded that of their reference Portland cement.Secondly, as shown in Figure 2, in blended cements where Ca(OH)_2_ concentration was extremely low, i.e., 40/60 and 30/70 percentages, an increase in compressive mechanical strengths for PY6/M blends was observed. This phenomenon can be explained by the higher C_3_S content (%) of this cement, which was 79.43%, in contrast, P1 contained 51.05%. For this reason, this PY6 blended cements were able to release a greater amount of Ca(OH)_2_ during cement hydration to chemically react with the SiO_2_^r−^ and Al_2_O_3_^r−^ of the CC to form a greater amount of several hydrated calcium aluminates [11] and CSH gels, later giving way to tobermorites [31]. In case of the results obtained from flexural strengths, the behavior could be explained in a similar way: the higher available amount of Ca(OH)_2_ in the mixing water and in storage water of the standard mortar specimens of blended cements [30] could react with the coarse and fine grains of the siliceous sand of mortar. So, the interfacial transition zone (ITZ) of siliceous aggregate–cement paste [32] increases its thickness with the age and, consequently, increases the pure tensile stress that must be carried out to break it. The quantification of this tensile stress could be carried out by the subjecting of mortar specimens to a flexural test, as has been done.Thirdly, with respect to the results obtained for the P/M 30/70 + CH blended cements when they were provided with an additional 7% SO_3_ content in form of natural gypsum stone, a great improvement in the compression mechanical strengths was observed in comparison with the corresponding P/M 30/70 + CH blends. The explanation for this good behavior can be explained by the excess of gypsum effect (=7.0% SO_3_) which, instead of reacting as an aggressive material, acts as a “setting controller” in cements blended with CC. Indeed, mechanical performance of such blends were of an order of magnitude like that observed in P/M 50/50 blended cements, that their clinker content was twenty percent higher (ett-rf and ett-lf formed in this case [33] naturally played a role in these beneficial results). With the rise in the SO_3_/Al_2_O_3_ molar ratio occasioned by the excess gypsum, the AFt did not convert into AFm. Both phases were detected by XRD, which will be shown later.

### 3.3. Le Chatelier–Anstett (L–A) Test

The sulfate resistance by the L-A test was evaluated for the same Portland and blended cements repaired to evaluate their mechanical behaviour

The L–A test findings (Figure 3) showed that, according to the criteria proposed by R. Talero [9,11,12,24] (ΔØ_28days_ ≤ 1.25% high SR; 1.25% < ΔØ_28days_ < 4.00% moderate SR; ΔØ_28days_ ≥ 4.00% low SR], PY6 was extraordinarily sulfate resistant, due to the small expansion of its specimen, ΔØ_28d_ was less than the specification, 1.25%. This result is totally dependent on the crystalline composition of this PY6 cement, as C_3_A content was 0.00% and C_4_AF content was 10.19% wt., (Table 1). In contrast, most PY6/M blends could be designed as non-sulfate-resistant cements [34,35,36], as the ΔØ_28d_ found for their L–A specimens was higher than the established specification. Only two blended cements, PY6/M 30/70 and PY6/M 20/80, could be considered as moderate SR.

In contrast, P1 couldn’t be qualified in the same category as PY6, as the results of the LA test confirmed that the 14.11% C_3_A content was significantly higher than the 5% specification defined for sulfate-resistant cement [25]. So, a more unfavorable behavior was observed for the blended cements prepared with P1 and M (Figure 3). However, it is significant in the case of blended cements P1/M 30/70 and P1/M 30/70 + CH when 7% SO_3_ was added. These cements become SR.

By way of summary, most of cements tested, couldn’t resist gypsum attack satisfactorily. Worse, however, was that the same behavior was recorded for all the P1/M, whose damage was most important with a rising proportion of M. Note that, even initially in PY6, sulfate resistance fell as the proportion of M in the blends increases. In general, the gypsum attack affected blended cements more quickly than P1 [11,12]. However, it is also important to note that, when a percentage of cement substitution is reached, the specimens expansion by the attack of the sulfates is reduced. This substitution ratio changes with the reference cement used.

Another very useful parameter that provides information every P/M blended cement behavior when exposed to sulfate attack was the diameter growth rate (V_CØ_/age). This parameter is graphed in Figure 4.

After studying the results obtained in the L-A test (Figure 3 and Figure 4), it is observed that, in general, ΔØ parameter was dependent on the high amount of CC added in blended cements. Therefore, the performance of these in a sulfate environment was a direct consequence derived from the physical substitution of Portland cement by M calcined clay. That is to say, the Al_2_O_3_^r−^ content present in M calcined clay, when converted in rapid formation ettringite (ett-rf), prevailed [34,35,36]. For this reason, ett-rf must be necessarily the main cause of the very fast expansion of the most L-A specimens. However, for two Portland cements P1 and PY6, the slow formation ettringite (ett-lf) from C_3_A and/or very slow formation ettringite (ett-vlf) from C_4_AF present prevailed [34,35,36]. So, all hydrated calcium aluminates that originated from M calcined clay at the beginning of cement hydration process are rapidly and totally transformed into ett-rf when L-A specimens were exposed to gypsum attack.

In this sense, it´s important remember that all ettringites are expansive despite their possible different origins, among others, from CC. There is direct relation between expansion and C_3_A content present in Portland cements and between expansion and Al_2_O_3_^r−^ content in CC, that was 15% (Table 1). The reactivity of Al_2_O_3_^r−^ of M calcined clay to form ettringite was notably higher than C_3_A from P1 and, in addition, it formed a greater quantity. For this, it can be rightly said that such a notable reactivity of Al_2_O_3_^r−^ could be found in the ideal physical-chemical state for this process [4].

This hypothesis was confirmed by the V_CØ_ (%) calculation (Figure 4). This parameter always dropped very faster for both families of blended cements than for its respective plain Portland cement and faster yet as the M calcined clay content increased up to a limit value. The decline began after the maximum value, that was always reached at 1 day.

#### Observations

There are several results that need to be analyzed separately for P1 blended cements:Firstly, the expansion of L-A specimens was most important with a rising proportion of M, but when this quantity of M calcined clay was higher than 60%, there was a significant decrease in L-A specimen expansion. It is common that, at the age of 1 day, ΔØ values are extremely high. Thus, ΔØ_1day_ values fit the following inequality:ΔØ1day: P1 < P1/M 30/70 < P1/M 50/50 < P1/ 40/60 < P1/M 40/60 + CH < P1/M 30/70 + CH

This behavior can be explained by this way: when M calcined clay was mixed with a Portland cement with C_3_A high content, P1, the expansive synergic effect (ESE) between them caused the rapid and early formation of ett-rf and the consequent increase in LA specimen’s diameter because of M was the main source of reactive alumina (Al_2_O_3_^r–^). This is because the high and speedy pozzolanicity generated by Al_2_O_3_^r–^ before the end of the first day in turn induced faster indirect sulfate-mediated hydration [11], especially of the high content of C_3_A with which it was mixed. However, this behavior, in high replacement cements, like CEM IV and CEM V [25], will only occur when there´s a minimum content of Ca(OH)_2_ in cement paste liquid phase that ensures the formation of the ett-rf. In this way, as the content of P1 cement in blended cements decreases, there will be less Ca(OH)_2_ available in cement liquid phase to react with the Al_2_O_3_^r−^ of the CC. This is the principal reason that P1/M 30/70 was the blended cement that less expanded. A similar behavior was found in mechanical strength results.

And to confirm the veracity the aforementioned hypothesis the performance of P1/M 40/60 + CH and P1/M 30/70 + CH must be observed. The L-A test specimens of these two blended cements were the ones with the highest expansion values. This behavior was due to two fundamental reasons:

(a) These two blended cements had the highest Al_2_O_3_^r−^ content of CC origin, namely 9% and 10.5%, respectively.

(b) Through the additional contribution of CH to these blended cements, it was possible to increase the content of Ca(OH)_2_ in the cement paste liquid phase.

In these cases, ett-rf causes a much larger diameter increase in L-A specimen’s diameter. Moreover, it occurs because there is enough content of Ca(OH)_2_ in their cement paste liquid phase that ensures the formation of the ett-rf. The XRD results (Figure 5) confirmed the behavior described earlier, with an additional amount of CH in L–A specimens, that promotes a higher amount of ett-rf formed in every cement paste. In other words, this particular type of ettringite, sourced from the Al_2_O_3_^r−^ content (8% and 9.5%) of CC, formed especially in the early stages of the test [11] when there exists enough portlandite in the liquid phase of cement pastes.

In short, P1/M 40/60 + CH and P1/M 30/70 + CH cases confirm that, the CC generated previously enough pozzolanic activity, a totally amount of ettringite from the C_3_A in P1 would be ett-rf only. This process would continue until the pozzolanic activity of CC was depleted due to the poor level of portlandite in the cement paste liquid phase. By that time, all the C_3_A may have been converted into ett-rf, and if not, any remaining C_3_A would be transformed into ett-lf [11].

2.Secondly, the insignificant expansion of L-A specimens in case of blended cements P1/M 30/70 and P1/M 30/70 + CH when previously a 7% SO_3_. The L–A test findings (Figure 3 and Figure 4) showed that, by the criteria put forward by Talero [9], these cement pastes were highly sulfate resistant, for their, ΔØ_28d_ were lower than the 1.25% ceiling. However, these results weren´t consistent with the crystalline composition of this P1 and the Al_2_O_3_^r−^ amount of CC, see Table 1. Therefore, the key question is why these L-A specimens didn´t expand as they should in this accelerated test.

The explanation for this significant behavior is found in the 7% additional amount of SO_3_ in the cement pastes in their previous hydration (14 days) before starting the L-A test. This additional content of gypsum in cement pastes plays a role like a mechanical strengths case: the excess gypsum reacts very fast as setting regulator in prehydration first stage (see Section 2.2. for an explanation of the L-A procedure). During this previous hydration cement paste treatment, an expansive synergic effect (ESE) [11] had taken place with non-expansion practical implications according to L-A physical parameters.

In short, the findings for ΔØ y V_CØ_ parameters prove the existence of a synergic effect because, the actual residual amount of the 15.05% (=7.0% SO_3_) gypsum added to the P1/M 30/70 and P1/M 30/70 + CH at the beginning of the test, produced a high ett-rf amount in both cement pastes that played a role in these beneficial results of sulfate durability. With the rise in the SO_3_/Al_2_O_3_ molar ratio occasioned by the excess gypsum, the AFt did not convert into AFm before the sulfate attack. Then, at the sulfate attack moment, there wasn´t reactive alumina available to form ett-rf, since it previously reacted with the excess sulfate of blended cements to form ettringite that didn’t have haFMSul consequences in the durability test.

There are several results that need to be analyzed separately for PY6 blended cements too:Firstly, the expansion of L-A specimens was most important with rising proportion of M, but a kinetic difference was found. When this quantity of M calcined clay was higher than 60% there was a significant decrease in L-A specimen expansion as well.

ΔØ_1day_ and ΔØ_28days_ values fit the following inequality which varies between them:

ΔØ_1day_: PY6 < PY6/M 20/80 = PY6/M 30/70 < PY6/M 30/70 + CH < PY6/ 40/60 = PY6/M 40/60 + CH < PY6/M 20/80 + CH < PY6/M 50/50

ΔØ_28days_: PY6 < PY6/M 20/80 = PY6/M 30/70 < PY6/M 30/70 + CH < PY6/ 40/60 < PY6/M 20/80 + CH < PY6/M 40/60 + CH = PY6/M 50/50

This behavior can be explained by different mode than P1 blended cements because the increase in L-A specimen’s diameter is only induced by ett-rf of Al_2_O_3_^r−^ of CC origin. In this sense it´s important to remember that PY6 was a SR Portland cement (≈0% C_3_A content). So, in the case of PY6/M 40/60 + CH blended cement, despite the high and speedy pozzolanicity generated by Al_2_O_3_^r−^ of CC, there couldn´t be as important a P1 blended cement ESE phenomenon because this specimen continued expanding after the first day. This will occur when there´s a minimum content of Ca(OH)_2_ in cement paste liquid phase because, when the quantity of CC was higher than 50%, the specimen´s expansion decreased as well as P1 blended cements.

To confirm this hypothesis again, the XRD results patterns for PY6/M 50/50 and 40/60 L-A specimens at 28-days are showed in Figure 6. An additional amount of CH in L–A specimens promotes a higher amount of ett-rf formed in every cement paste. This ettringite, sourced from the Al_2_O_3_^r−^ content (8% and 9.5%) of CC, is formed especially in the early stages of the test [9] when there exists enough portlandite in the liquid phase of cement pastes.

## 4. Conclusions

The conclusions drawn from the findings discussed in this paper are summarized in the following.

The obtained results confirm the SR increase and mechanical strengths decrease of a Portland cement when it´s replaced by a CC in amounts greater than 40%. This result is a consequence of the chemical effect derived from the pozzolanic activity of the CC which was blended. A considerable decrease in portlandite was then caused in the liquid phase of its paste.With such high CC replacement percentages, various blended cements met the regulated specifications for compressive mechanical strengths (CMS) and flexural mechanical strengths (FMS). In contrast, some blended cements didn’t reach the minimum available amount of portlandite in the liquid phase so that CC could develop its full pozzolanic activity to guarantee the reaction products stability.In sulfate resistance tests, the expansion decrease ratio in the cements with a higher quantity of CC is due to the lower proportional concentration of portlandite from the beginning to the end of the test. This lower amount of CH prevented the formation of all the ett-rf amounts of CC origin. In the case of P1 cement, the expansion was also induced by the content of C_3_A (%) of P1, which was able to stimulate CCWith the external addition of CH, the CC was able to show its true chemical character and behavior against the attack of sulfates. With this, the greater amount of ett-rf formation of Al_2_O_3_^r–^ pozzolan origin caused a decrease in its RS and the increase in mechanical strengths was generated for 28 and 90 days in EN 196-1 mortar specimens of the new pozzolanic cement CEM IV/B type.

## Figures and Tables

**Figure 1 materials-13-04206-f001:**
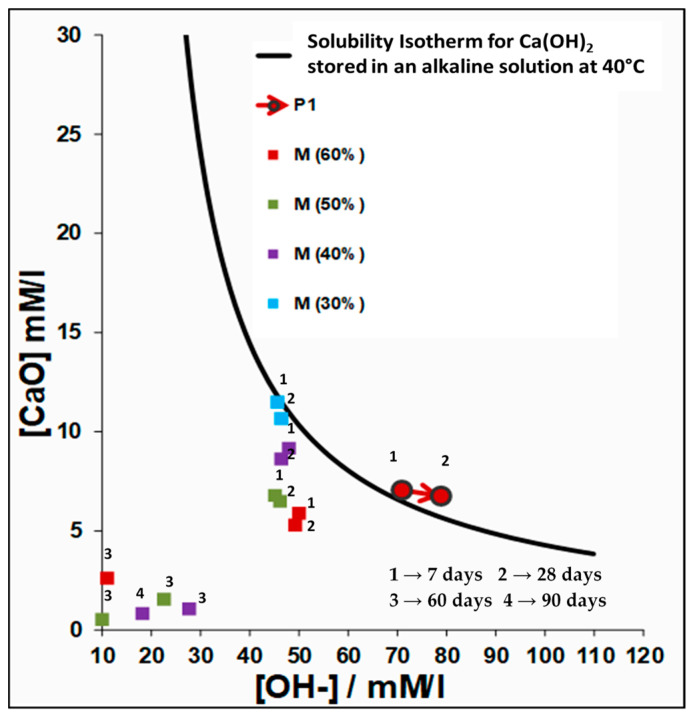
Frattini test results for P1 with and without calcined clay M.

**Figure 2 materials-13-04206-f002:**
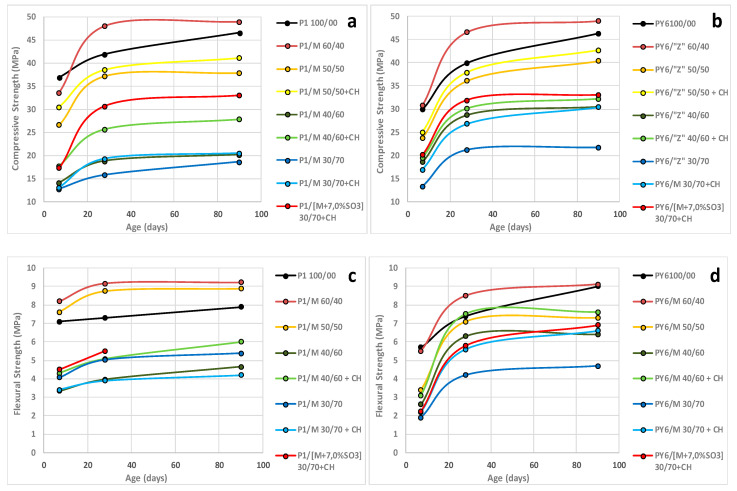
Flexural and Compressive strengths of P1, PY6 and their blended cements. (**a**,**c**) P1/M. (**b**,**d**) PY6/M.

**Figure 3 materials-13-04206-f003:**
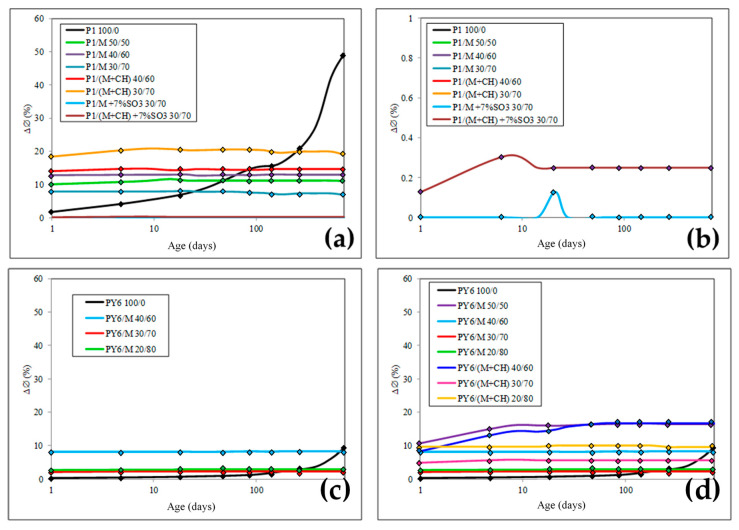
L–A specimens: increase in diameter, ΔØ (%), versus age. (**a**,**b**) P1 and their M blended cements, (**c**,**d**) PY6 and their M blended cements.

**Figure 4 materials-13-04206-f004:**
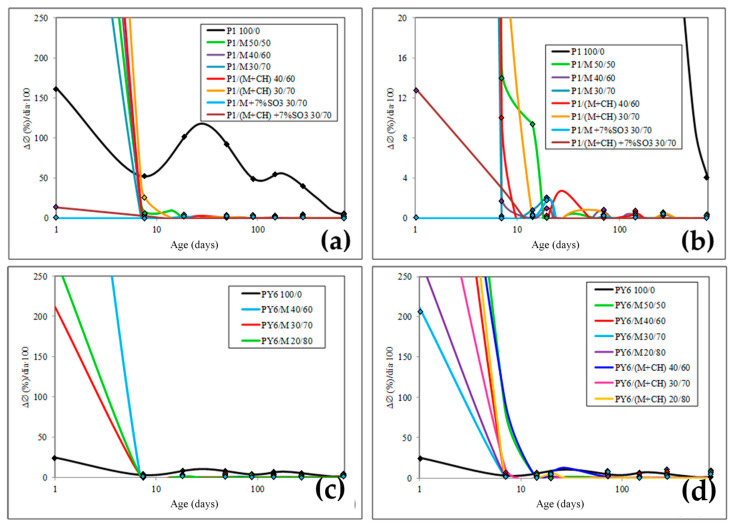
L–A specimens: diameter growth rate, V_CØ_ (%), versus age. (**a**,**b**) P1 and their M blended cements, (**c**,**d**) PY6 and their M blended cements.

**Figure 5 materials-13-04206-f005:**
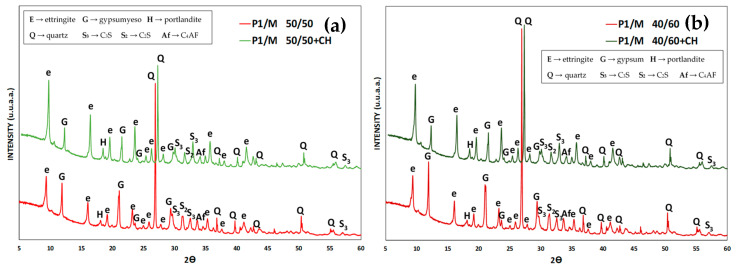
XRD patterns. (**a**) P1/M 50/50 and P1/M 50/50 + CH. (**b**) P1/M 40/60 and P1/M 40/60 + CH. L-A specimens at 28-days.

**Figure 6 materials-13-04206-f006:**
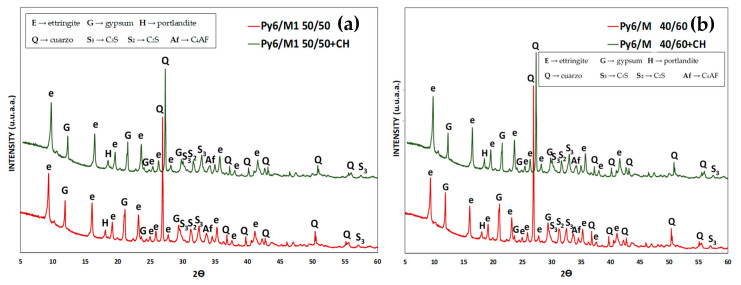
XRD patterns. (**a**) P1/M 50/50 and PY6/M 50/50 + CH. (**b**) P1/M 40/60 and PY6/M 40/60 + CH. L-A specimens at 28-days.

**Table 1 materials-13-04206-t001:** Physics-Chemicals determinations of materials.

Chemical Parameters (%)	Portland Cements		Calcined Clay (M)
	P1	PY6	
L.O.I.	1.60	1.11	0.60
I.R.	0.70	0.15	0.22
SiO_2_	19.18	21.70	73.55
Al_2_O_3_	6.44	1.52	23.11
Fe_2_O_3_	1.75	4.11	1.19
CaO	63.94	67.97	0.63
MgO	1.48	0.42	0.03
Na_2_O	0.90	0.43	0.07
K_2_O	0.52	0.20	0.70
TiO_2_	-	-	
PY6O_5_	-	-	
SO_3_	3.50	2.34	
SiO_2_^r−^			38.00
Al_2_O_3_^r−^			15.00
TOTAL	100.01	99.50	100.10
H_2_O (105 °C)	0.24	0.22	0.00
Free CaO	1.9	1.75	-
Blaine Specific Surface (BSS) (m^2^/kg)	319	329	726
Specific Density (Kg/m^3^)	3080	3210	2550
C_3_S	51.05	79.43	
C_2_S	16.48	2.29	
C_3_A	14.11	0.00	
C_4_AF(+C_2_F)	5.33	10.19	

**Table 2 materials-13-04206-t002:** Flexural and compressive strength results.

Age (days)		P1 100/00	P1/M 60/40	P1/M 50/50	P1/M 50/50 + CH	P1/M 40/60	P1/M 40/60 + CH	P1/”Z” 30/70		P1/”Z” 30/70 + CH	
								M	M + 7.0%SO_3_	M	M + 7.0%SO_3_
7	Compresive (MPa)	36.9	33.6	26.7	30.6	14.2	17.8	12.8		13.1	17.3
	Flexural (MPa)	7.1	8.2	7.6		3.3	4.3	4.1		3.4	4.5
28	Compresive (MPa)	41.9	48.0	37.2	38.6	18.9	25.7	15.8		19.3	30.7
	Flexural (MPa)	7.3	9.2	8.8		4.0	5.1	5.0		3.9	5.5
90	Compresive (MPa)	46.6	48.9	37.9	41.1	20.2	27.9	18.6	26.0	20.5	33.0
	Flexural (MPa)	7.9	9.2	8.9		4.7	6.0	5.4		4.2	0.0
**Age (days)**		**PY6 100/00**	**PY6/M 60/40**	**PY6/M 50/50**	**PY6/M 50/50 + CH**	**PY6/M 40/60**	**PY6/M 40/60 + CH**	**PY6/”Z” 30/70**		**PY6/”Z” 30/70 + CH**	
								**M**	**M + 7.0%SO_3_**	**M**	**M + 7.0%SO_3_**
7	Compresive (MPa)	29.9	30.7	23.7	25.0	18.5	19.5	13.3		16.9	20.1
	Flexural (MPa)	5.7	5.5	3.4		2.6	3.1	1.9		2.2	2.3
28	Compresive (MPa)	39.9	46.6	36.1	37.9	28.7	30.1	21.2		26.8	31.9
	Flexural (MPa)	7.4	8.5	7.1		6.3	7.5	4.2		5.6	5.8
90	Compresive (MPa)	46.3	49.0	40.4	42.7	30.5	32.2	21.7	32.0	30.4	33.0
	Flexural (MPa)	9.0	9.1	7.3		6.4	7.6	4.7		6.6	6.9

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
