# Peer review of "Portland Cements with High Content of Calcined Clay: Mechanical Strength Behaviour and Sulfate Durability"

_materials, 2020, doi:10.3390/ma13184206_

Round 1
Reviewer 1 Report
This manuscript presents a study on some mechanical properties and on sulfate resistance of blended cements. The manuscript is generally well written, but there are some aspects that need to be corrected before publication. A list of comments/suggestions follows:
- Page 1, lines 9-13, especially line 13. This sentence does not have a verb: “In the same way, the mechanical strengths of these blended cements”. However, the text between lines 9 to 13 should be modified because the first aspect covered in the manuscript is the mechanical properties and only after this comes the sulphate resistance. Since the order in the abstract is inverse, it should be changed.
- Page 5, Lines 174-177. Before this paragraph, the authors need to give more information on the results which led to Fig. 2. This should include the number of tests and the values obtained at each age of concrete, for instance. Without such information it is not possible to have an idea of the variability of the results and of the approach for fitting the results to the drawn curves.
- The criticism on the lack of information prior to figure 2 is also valid for Figures 3 and 4.
- Find and correct “sulfatic”.
Author Response
All suggestions have been considered. All proposed corrections have been made. We thank the evaluator for this kind suggestions.Reviewer 2 Report
The literature review is unacceptable. The authors, out of 24 items (excluding standards), cite 15 of their own works, and the vast majority of them covering the period a few years ago. It seems that no one else has been publishing more about Portland cements – which is not true. Literature should be completed and I suggest that self-citations should not exceed 1/3 of all literature references. Below are some suggestions to consider:
- Materials 12 (12), (2019), 1942
- Cement and Concrete Composites, vol. 113 (2020), 103710
- Construction and Building Materials, vol. 250, (2020), 118884
Detailed comments
- Figure 1 is illegible. It is not known which results correspond to 7 days, 28days , 60 days and 90 days. What does the trend line mean? Which results is it supposed to approximate? What determines the obtained results - days or percentages? It seems unnecessary in this Figure.
- Information about the measurement error is missing. Are the results presented, for example, in Figures 2-4 the averages of the measurements? How many test samples were there? What was the spread of the results?
- In line 344, the authors say that they have confirmed the hypothesis. Are you sure this is a hypothesis? A hypothesis is formed when for certain facts there are no available published accounts and checks. Is it like that in that case?
Author Response
Firstly, six additional references have been incluided. In addition, three own references have been removed.
Secondly, respect to Figure 1: This figure shows the results of a standardized test on cement research. Frattini test according to EN 196-5.
This test determines the amount of Ca(OH)2 originated from cement hydration that taking place in an aqueous solution containing the sample. These [OH-] and [CaO] results are compared with the calcium hydroxide [Ca(OH)2] solubility isotherm curve estimated for an alkaline solution at 40°C. Those points underneath that curve (subsaturation zone) are indicative the pozzolanic activity of the material that the cement was blended.
In addition, figura has been modified to facility its comprehension.
Trirdly, respect to Figures 2-4:
. Fig. 2 has been modified to facility its comprehension.
. Fig. 3 and 4. are the graphic representation of the expansion data given by the L-A specimens of each blended cement prepared at the specified ages. A specimen of each belendedcement was prepared according to the test procedure.
Thanks for your kindly suggestions
Reviewer 3 Report
The manuscript presents a study on mechanical strength behavior and durability of Portland cements with high contents of calcined clay. The manuscript is well written and has the potential to contribute to this domain. However, the authors can consider my following comments to improve the manuscript:
1. Title: The title is incorrectly written. Please check and modify accordingly.
2. Abstract: Please include the full form of abbreviations (e.g. CEM).
3. Keywords: Include the most relevant keywords (e.g. 'low carbon' can be replaced).
4. Introduction: Please include some more relevant and latest citations.
5. Materials and Methods: Modify Table 1. Check the caption, spacing and presentation
6. Modify Fig.1 with a proper resolution, font size and thinner curves.
7. Add a proper explanation to the 'P1/[M+7,0%SO3]
30/70+CH' curve in Fig. 2(c).
8. Numberings, uniformity in captions, font size should be checked and corrected for all figures (e.g. Fig 3, Fig.4: numberings patterns are not consistent; Fig. 5: clarity, headings & fonts)
9. Conclusion: Should be significantly concise. Include unique and specific outcomes only.
Author Response
All reviewer suggestions have been considered. All changes suggested by the reviewer have been made. thanks for your helpReviewer 4 Report
Dear Authors,
The aim of this study to evaluate mechanical strength of Portland cements with high contents of calcined clay. While the topic is fitting to the journal scope, some concerns were raised to publish as a scientific paper. Revise the manuscript by following comments.
Major points
Figure 3, Figure 4, and Figure 5 were not results of mechanical strength. The title of this study must be rephrased.
There was no clear purpose in the Abstract section and the Introduction section. Add the clear purpose in both sections.
Where is the points of 7 days, 28 days, 60 days, and 90 days? It was difficult to read. Revise them.
There was no explanation about how they measured flexural strength and compressive strength. Add them in the Materials and Methods section.
Page 5, Line 185
There was no explanation about statistical analysis. How they confirmed the significant decrease? Revise them.
Why the red curve in Figure 2c was disappeared around 30 days?
The conclusion section was too long. Summarize them.
Minor points
Page 1, Line 10
CEM, A-SR, B-SR must be spelled out at the first use in each section. Make sure all related points and revise them.
Page 1, Line 13
This is not sentence. Revise it.
Page 1, Line 18
"sulfatic" should be modified to "sulfate".
Table 1
"Gravity" should be modified to "Density". Also, SI unit should be used. (kg -> N).
Page 5, Line 209
What was "<>"? Typo?
Page 10, Line 452-453
Why the short term of compressive mechanical strength was RMC? Same for RMF.
Author Response
All reviewer suggestions have been considered.
All changes suggested by the reviewer have been made.
Thanks for your help
Round 2
Reviewer 1 Report
Comments 2 and 3 of my first review work were not conveniently addressed by the authors in this new version:
- ...
- Page 5, Lines 174-177. Before this paragraph, the authors need to give more information on the results which led to Fig. 2. This should include the number of tests and the values obtained at each age of concrete, for instance. Without such information it is not possible to have an idea of the variability of the results and of the approach for fitting the results to the drawn curves.
- The criticism on the lack of information prior to figure 2 is also valid for Figures 3 and 4.
Author Response
Answers to the reviewer's kind comments are as follows:
- Page 5, Lines 174-177. Before this paragraph, the authors need to give more information on the results which led to Fig. 2. This should include the number of tests and the values obtained at each age of concrete, for instance. Without such information it is not possible to have an idea of the variability of the results and of the approach for fitting the results to the drawn curves.
These results have been obtained following the EN 196-1 stardard, so we consider that it isn´t necessary include the number of test and values obtained at each age. In this sense, the reader only has to look at reference number 24 to check the number of specimens that are manufactured according to the standard procedure.
EN 196-1 standard:
“The method includes the determination of the compression and flexural strengths of prismatic specimens of dimensions 40x40x160. The specimens (3) are manufactured with a plastic mortar, composed of one part by mass of cement, three parts by mass of CEN standardized sand and half part of water (ratio w/c 0.5)”.
“The specimens are kept in the mold in a humid atmosphere for 24 hours and, after debonding, the specimens are immersed in water until the moment of resistance tests. At the required age, the specimens are removed from their moist preservation medium, they are broken in flexion, the flexural strength determined, and each half is subjected to compression.”
“Flexural strength. Calculation and expression of the test results: The result of the flexural strength test is calculated as the arithmetic mean of the three individual results, expressed, each one of them, rounded to the nearest 0.1MPa and obtained from the determination made on the set of three test specimens.
“Compressive strength. Calculation and expression of the test results: The result of the compressive strength test is calculated as the arithmetic mean of the six individual results, expressed, each one, rounded to the nearest 0.1MPa and obtained from the six determinations made on the set of three specimens”.
Despite this, if the reviewer considers that we write this information in the paper, we will do this without any problem.
2. The criticism on the lack of information prior to figure 2 is also valid for Figures 3 and 4.
This point could be explained by the same way. These results have been obtained following the Le Chatelier-Anstett procedure, so we consider that it isn´t necessary include the number of test and values obtained at each age. In this sense, the reader only has to look at reference number 23 to check the number of specimens that are manufactured according to the standard procedure.
Despite this, if the reviewer considers that we write this information in the paper, we will do this without any problem.
However, figures 3 and 4 have been modified for a better understanding of the reader
Reviewer 2 Report
I accept the changesAuthor Response
Thank you very much
Best regards
Reviewer 3 Report
I recommend publication
Author Response
Thank you very much
Best regards
Reviewer 4 Report
Dear Authors,
Unfortunately, the manuscript was not revised appropriately by following the reviewer's comments. Some of major comments and minor comments were not revised and were still remaining.
Ex.
There was no clear purpose in the Abstract section and the Introduction section.
There was no explanation about how they measured flexural strength and compressive strength.
There was no explanation about statistical analysis. How they confirmed the significant decrease? Revise them.
The conclusion section was too long.
CEM, A-SR, B-SR must be spelled out at the first use in each section.
Author Response
Answers to the reviewer's kind comments are as follows:
- There was no clear purpose in the Abstract section and the Introduction section.
Abstract and introduction sections have been modified to make clearer the purpose of them
- There was no explanation about how they measured flexural strength and compressive strength.
These results have been obtained following the EN 196-1 standard, so we consider that it isn´t necessary include the number of test and values obtained at each age. In this sense, the reader only has to look at reference number 24 to check the number of specimens that are manufactured according to the standard procedure.
EN 196-1 standard:
“The method includes the determination of the compression and flexural strengths of prismatic specimens of dimensions 40x40x160. The specimens (3) are manufactured with a plastic mortar, composed of one part by mass of cement, three parts by mass of CEN standardized sand and half part of water (ratio w/c 0.5)”.
“The specimens are kept in the mold in a humid atmosphere for 24 hours and, after debonding, the specimens are immersed in water until the moment of resistance tests. At the required age, the specimens are removed from their moist preservation medium, they are broken in flexion, the flexural strength determined, and each half is subjected to compression.”
“Flexural strength. Calculation and expression of the test results: The result of the flexural strength test is calculated as the arithmetic mean of the three individual results, expressed, each one of them, rounded to the nearest 0.1MPa and obtained from the determination made on the set of three test specimens.
“Compressive strength. Calculation and expression of the test results: The result of the compressive strength test is calculated as the arithmetic mean of the six individual results, expressed, each one, rounded to the nearest 0.1MPa and obtained from the six determinations made on the set of three specimens”.
Despite this, if the reviewer considers that we write this information in the paper, we will do this without any problem.
- There was no explanation about statistical analysis. How they confirmed the significant decrease? Revise them.
The statistical analysis made was according to EN 196-1 standard, so we consider that it isn´t necessary include it because the reader can find it in section 10.2 of reference number 24.
- The conclusion section was too long.
The conclusion section has been reduced.
- CEM, A-SR, B-SR must be spelled out at the first use in each section.
This denomination isn´t own. This is international denomination of different type of cements according to European EN 197-1 standard. The meaning of this type of cements is written in number 30 reference.
Round 3
Reviewer 1 Report
The manuscript continues not responding the comments sent previously.
For instance:
- Table 2 has a label "Age (days)", but this label is not aligned with the numbers 7, 28, 90. As it is, it is not clear that the compressive tests were carried out in the same days.
- It is not sufficient to say that the tests were carried out according with given standards. The authors must include the number of tests at each age because the standard does not prevent you to do more tests than the minimum number and this is research, not a construction on site. In any case, in research we tend to look to average values rather than to the characteristic strength (most appropriate to ensure structural safety of the constructions).
- The variability of the tests are not shown in any of the graphs.
This is important for the quality of the manuscript.
Author Response
Table 2 has been modified to shows the information more clearly
Materials and methods section has been modified to define more information about these practices. The number of tests at each age and other information on data processing have been included in this review manuscript.
We hope that these changes in the article are to the liking of the reviewer
Reviewer 4 Report
Dear Authors,
The manuscript was mostly revised by following the reviewer's comments. Please make sure the following comments to improve readability of the manuscript.
Minor point
Abstract and Introduction
Please describe clear purpose of this study in the Abstract and the Introduction section.
Ex. The aim of this study was to evaluate mechanical strengths and sulphate resistance of blended cements with high content of calcined clay as a pozzolanic addition.
The readers may not familiar with EN 197-1. Even to follow EN 197-1, specimen dimension and a number of specimen should be described at least. Same for the statistical analysis.
Hope you will get acceptance soon.
Author Response
Abstract and introduction sections have been modified to define the purpose of this study clearly.
Materials and methods section has been modified to define more information about thes practises.
Thank you very much for your kindly suggestions
Round 4
Reviewer 1 Report
It is important to present the maximum information in a manuscript. I feel that even more information could be added, but now a minimum level was reached. Therefore, in my opinion, the manuscript can now be accepted.